# Risk factors for gallstone disease onset in Japan: Findings from the Shizuoka Study, a population-based cohort study

Kazuya Higashizono[1,2]* , Eiji Nakatani[1] , Philip Hawke[3], Shuhei Fujimoto[1], Noriyuki Oba[2]

1 Graduate School of Public Health, Shizuoka Graduate University of Public Health, Shizuoka, Japan, 2 Department of Gastrointestinal Surgery, Shizuoka General Hospital, Shizuoka, Japan, 3 School of Pharmaceutical Sciences, University of Shizuoka, Shizuoka, Japan

☯ These authors contributed equally to this work.
* toen0108@gmail.com

## Abstract

In the research literature on factors associated with gallstones, large population-based cohort studies are rare. We carried out a study of this type to explore risk factors for the onset of gallstones. This study included Japanese participants aged 40–107 years who were followed prospectively from January 2012 to September 2020 using a dataset composed of two individually linked databases, one containing annual health checkup records and the other containing medical claims for beneficiaries of the National Health Insurance System and the Medical Care System for Elderly in the Latter Stage of Life in Shizuoka Prefecture, Japan. Among the 611,930 participants in the analysis set, 23,843 (3.9%) were diagnosed with gallstones during the observational period (median [max]: 5.68 [7.5] years). Multivariate analysis revealed that risk of gallstone disease was increased by male sex, cerebrovascular disease, any malignancy, dementia, rheumatic disease, chronic pulmonary disease, hypertension, and *H. pylori*-infected gastritis. These findings provide essential insights into the etiology of cholelithiasis and may contribute to efforts to reduce the incidence of the disease.

**Data Availability Statement:** According to Shizuoka Prefecture's data use agreement with local insurers, readers cannot access the analyzed data. Researchers interested in accessing this data

## Introduction

Gallstone disease, the condition of stones or sludge developing in the gallbladder or bile duct, is one of the most common gastrointestinal diseases [1]. Gallstones are associated with potential risk of cholecystitis, pancreatitis, biliary tract obstruction, and gallbladder cancer [2]. Gallstone-associated diseases often require cholecystectomy or percutaneous/endoscopic biliary drainage due to late diagnosis and treatment of severe acute biliary inflammation/infection associated with sepsis and multiple organ failure. The reported mortality of acute cholangitis varies from 2.5% to 65% [3]. Therefore, elucidation of the risk factors for gallstones may help to prevent the onset of cholangitis, leading to increased life expectancy, reduction of medical costs, and improved social productivity.

set may apply to Shizuoka Prefecture to request access. Please contact the staff of Shizuoka Graduate University of Public Health (Email: info@s-sph.ac.jp).

**Funding:** The Shizuoka Graduate University of Public Health conducts research projects on public health under contract to Shizuoka Prefecture, including this study.

**Competing interests:** The author have declared that no competing interests exist.

Gallstones are composed of a mixture of cholesterol, calcium salts of bilirubinate or palmitate, proteins, and mucin. Based upon their predominant constituents, gallstones are broadly classified into cholesterol, brown pigment, and black pigment stones [4]. A variety of risk factors have been reported to be associated with gallstone formation. Established risk factors for the development of cholesterol gallstones due to enhanced cholesterol synthesis and secretion include genetic background and lifestyle, as well as internal disorders such as aging, female sex, pregnancy, obesity, rapid weight loss, diabetes mellitus, and dyslipidemia [5–10]. Predominant risk factors for pigment stones include liver dysfunction [11], Crohn's disease [12], and hyperbilirubinemia due to underlying genetic predisposition [13]. Spinal cord injury [14], prolonged fasting / parenteral nutrition [15], and gastrectomy [16] have also been reported to be risks for gallstone onset. Clinically, the incidence of gallstone disease has been increasing in the past decade in East Asia, coincident with increasing calorie and fat consumption, decreasing fiber intake, and increasing prevalence of sedentary lifestyle [17].

Despite the large number of previous epidemiological studies analyzing background factors related to gallstones, there have been few large-scale epidemiological studies, especially in East Asian countries. The aim of this study was to identify factors associated with the onset of gallstone disease in the healthy Japanese population, which can be assumed to reflect the characteristics of East Asians in general.

## Methods

### Data source

The study was based on the Shizuoka Kokuho Database, which is derived from a database providing the linked data of Federation of National Health Insurance Association (FNHIA) subscribers, including demographic and registration data, medical claims data, and health checkup data of enrollees in Shizuoka Prefecture [18]. Shizuoka Prefecture has a population of about 3.6 million people with climatic conditions and population distribution representative of Japan. The database represents 25% of the population under 65 and 75% of the population over 65 in the Prefecture, mainly due to it including individuals enrolled in the country's national health insurance programs (aged < 75, approximately 20% of all prefectural residents) and beneficiaries of late-stage elderly healthcare (aged ≥ 75, all prefectural residents in the age group).

### Japanese medical insurance and health checkup systems

Japan's medical system is based on a comprehensive insurance system. There are two types of health insurance for people aged < 75 years: Employee Health Insurance for the employees of government organizations and large companies, and National Health Insurance for small business owners and their employees. Health insurance for people aged ≥ 75 years is provided by the Medical Care System for Elderly in the Latter Stage of Life.

The Japanese Ministry of Health, Labour and Welfare recommends annual health checkups for insured persons 40 year of age and older. These checkups focus especially on visceral fat obesity.

### Study design, data availability period, and study population

The study scheme is shown in Fig 1. The study was analyzed as a retrospective cohort generated from the database. The dataset comprised 8.5 years of longitudinal data from April 2012 to September 2020. All enrollees were investigated using the individually linked data in the databases for their annual health checkups and their insurance claims. Each enrollee's data

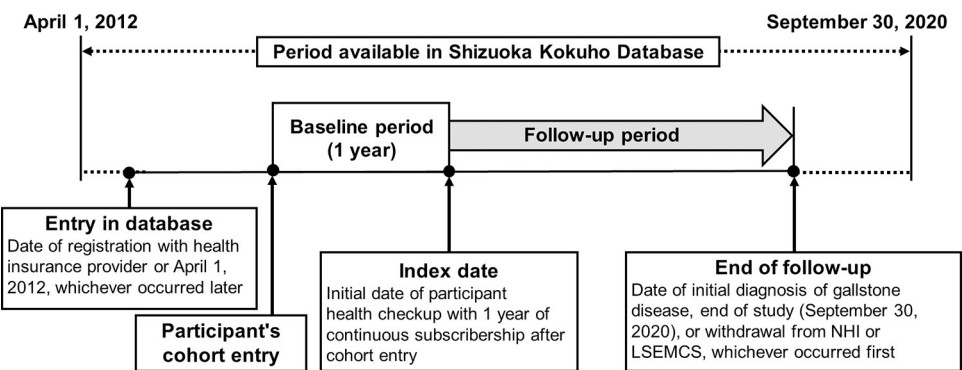

**Fig 1. Study schema.** Cohort entry is defined as the date of registration with the health insurance provider or April 1, 2012, whichever occurred later. The index date is defined as the initial date of annual health checkups with ≥ 1 year continuous subscribership to the health insurance system. "Follow-up period" is defined as the interval between the index date and (1) the end of the study (September 30, 2020) or (2) the withdrawal date from the health insurance system, whichever occurred first.

availability period was defined as the time from the date of insurance registration or April 2012, whichever was later, to the date of insurance withdrawal or September 2020, whichever was earlier.

We excluded participants who received a diagnosis of gallstones before the start date of their health checkup, those who did not undergo a health checkup, and those whose observation period was < 1 year.

## Onset of gallstone disease

Patients were classified as having gallstones when their insurance claims showed the code K80, which is listed in the International Classification of Diseases - 10th Revision (ICD-10) [19]. The claim codes for the medical interventions for the gallstone patients in this study are shown in S1 Table.

## Covariates

For the variable of specific comorbidity, the one-year period prior to the date of the medical examination was used as the search period, and the variable was judged as being present if the disease had been recorded as confirmed in the insurance claims data. Variables likely to be associated with the onset of gallstones were similarly defined based on the ICD-10 list, with a search period of one year used to determine their presence or absence.

All of the health checkup data required for the study, including age, gender, height, weight, systolic and diastolic blood pressure, smoking habits, as well as the use of antihypertensive, lipid-lowering, and hypoglycemic drugs, was available in the form of self-reported questionnaire data.

## Statistical analysis

The data was summarized as mean and standard deviation for continuous variables and as frequency and percentage for categorical variables. Univariable and multivariable Cox proportional hazards regression analyses were performed to explore factors associated with gallstone onset. Hazard ratio (HR), 95% confidence interval (CI), and P value were calculated, and a Wald test was conducted. Potential risk factors such as key etiological and epidemiological factors were examined using a regression analysis. To carry out the analysis conservatively, no

model selection was carried out, and all variables that reached statistical significance in the univariable regression analysis were entered into the multivariable regression analysis. Variables with a relatively large number of missing values in the health checkup data were excluded from the multivariable model. One of two variables with high correlation was not used in the multivariable model owing to multicollinearity, based on the criterion of an absolute Spearman's correlation coefficient of > 0.4. As the missing covariates did not occur completely at random among all participants, a simple missing data imputation was not carried out. A P value of < .05 was considered statistically significant. All analyses were performed using the software packages EZR Version 1.27 (Saitama Medical Center, Jichi Medical University, Tochigi, Japan) and SAS Version 9.4 (SAS Institute Inc., Cary, NC, USA).

## Ethics

All enrollee data was anonymized by the Federation of National Health Insurance Association to protect participant confidentiality [18]. This study adhered to the principles of the Declaration of Helsinki and was approved by the Medical Ethics Committee of Shizuoka Graduate School of Public Health in Shizuoka, Japan (#SGUPH_2021_001_040).

## Results

### Study population

Among the 2,654,567 individuals in the Shizuoka Kokuho Database, 741,009 underwent health checkups. Among these, 113,195 patients who had undergone an observation period of < 1 year and 15,884 who had already been diagnosed with gallstones were excluded. Data for the 611,930 remaining individuals was analyzed (Fig 2).

### Gallstone disease onset

During the observation period (median [max]: 5.68 [7.5] years), 23,843 patients (3.9%) were newly diagnosed with gallstones.

Among these, 4,759 (20.0%) had undergone cholecystectomy and percutaneous/endoscopic drainage. The baseline characteristics of individuals with gallstones who underwent treatment for the disease and those who did not are shown in S1 Table. The baseline characteristics for the gallstone and other patients are shown in Table 1.

### Identification of risk factors of gallstone disease onset

Potential risk factors and significant variables were evaluated using a univariable Cox regression analysis (Table 2). Several variables had correlations of > 0.4 (S3 Table). Variables with p<0.05 in the univariable analysis were entered into the multivariable model.

The multivariable regression analysis showed that increasing age and male sex (HR: 1.09 [95% CI: 1.06–1.12]) increased the risk of gallstones (Table 3). The presence of comorbidities such as cerebrovascular disease (1.14 [1.10–1.19]), any malignancy (1.30 [1.24–1.36]), dementia (1.13 [1.04–1.23]), rheumatic disease (1.23 [1.14–1.33]), liver disease (1.56 [1.16–2.10]), congestive heart failure (1.17 [1.12–1.23]), chronic pulmonary disease (1.15 [1.12–1.19]), diabetes (1.15 [1.08–1.22]), hypertension (1.10 [1.06–1.13]), and *H. pylori*-infected gastritis (1.25 [1.18–1.32]) were also risks for the disease. Further, the health checkup data revealed that higher values for BMI (1.04 [1.04–1.05] per 1 kg/m$^2$ increase), GGT (1.15 [1.13–1.16] for 100 U/L increase), HbA1c (1.03 [1.01–1.04] per 1% increase), and triglycerides (1.04 [1.02–1.06] per 100 mg/dL increase) were associated with the onset of the disease. However, a habit of walking or other physical exercise > 1 h / week (0.91 [0.94–0.89]), higher LDL cholesterol

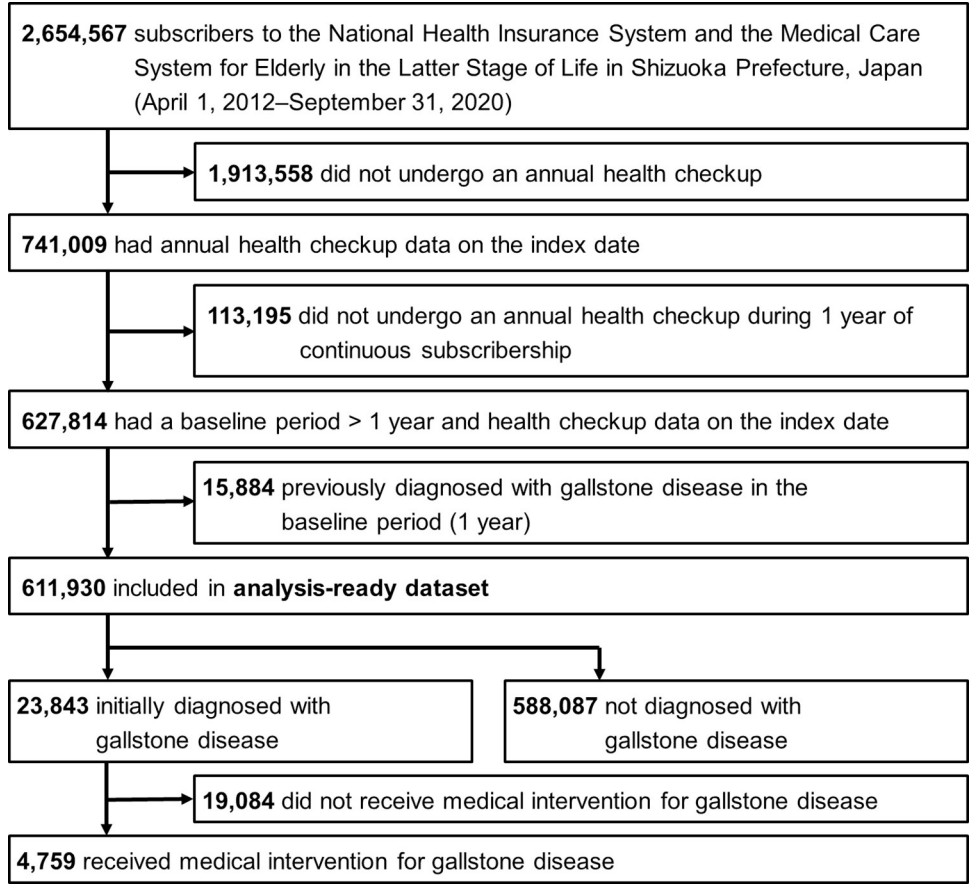

**Fig 2. Flow chart of study.**

(0.98 [0.98–0.99] per 10 mg/dL increase), and higher systolic blood pressure (0.98 [0.97–0.99] per 10 mmHg increase), were associated with reduced risk of gallstones.

## Discussion

The three main causes of gallstone genesis are enhanced cholesterol nucleation, impaired gallbladder emptying with stasis, and intestinal hypomotility. Enhanced cholesterol nucleation is the most common type, accounting for 75–80% of occurrences, and is caused by impaired cholesterol metabolism [20]. Thus, metabolic syndrome and associated diseases such as higher BMI, hypertension, and diabetes mellitus have been widely reported to play a pivotal role in the pathogenesis of gallstone disease. The results of the present study are consistent with these previous studies.

This study demonstrates that increasing age, male sex, a variety of comorbidities (cerebrovascular disease, any malignancy, rheumatic disease, liver disease, congestive heart failure, chronic pulmonary disease, renal disorder, diabetes, hypertension, and pylori-infected gastritis), and several medical checkup items (BMI, γ-GT, HbA1c [%]) were closely associated with development of gallstones. Several of these factors have been identified in previous studies [21–23]; however, the present study is the first large-scale cohort study to reveal the risk factors of rheumatic disease, malignancy, *H. pylori*-infected gastritis, chronic pulmonary disease, and renal disorder.

**Table 1. Participant characteristics and variables for univariable Cox model analysis.**

| Variable | Category or unit | Patients with gallstones | Other patients |
|---|---|---|---|
| | | N = 23843 | N = 588087 |
| Sex | Women | 12797 (53.7) | 335191 (57.0) |
| | Men | 11046 (46.3) | 252896 (43.0) |
| Age | 0 to <40 years | 41 (0.2) | 4699 (0.8) |
| | 40 to <50 years | 620 (2.6) | 44827 (7.6) |
| | 50 to <60 years | 1256 (5.3) | 52581 (8.9) |
| | 60 to <70 years | 7779 (32.6) | 222038 (37.8) |
| | 70 to <80 years | 8980 (37.7) | 176696 (30.0) |
| | 80 years to | 5167 (21.7) | 87246 (14.8) |
| **Comorbidities** | | | |
| Cerebrovascular disease | Presence | 4262 (17.9) | 73757 (12.5) |
| Any malignancy | Presence | 2386 (10.0) | 41148 (7.0) |
| Dementia | Presence | 753 (3.2) | 13749 (2.3) |
| Myocardial infarction | Presence | 457 (1.9) | 7897 (1.3) |
| Renal disease | Presence | 660 (2.8) | 11377 (1.9) |
| Rheumatic disease | Presence | 721 (3.0) | 13352 (2.3) |
| Liver disease | Presence | 47 (0.2) | 641 (0.1) |
| Congestive heart failure | Presence | 2708 (11.4) | 44666 (7.6) |
| Chronic pulmonary disease | Presence | 5258 (22.1) | 106453 (18.1) |
| Diabetes | Presence | 1555 (6.5) | 26978 (4.6) |
| Hypertension | Presence | 13425 (56.3) | 265039 (45.1) |
| Pylori-infected gastritis | Presence | 1513 (6.3) | 28545 (4.9) |
| History of gastrectomy | Presence | 18 (0.1) | 193 (0.0) |
| HIV infection | Presence | 1 (0.0) | 48 (0.0) |
| Crohn's disease | Presence | 16 (0.1) | 269 (0.0) |
| Spinal injury | Presence | 1 (0.0) | 18 (0.0) |
| Total parenteral nutrition | Presence | 0 (0.0) | 5 (0.0) |
| **Medical checkup** | | | |
| Frequency of alcohol consumption | Never | 12178 (62.4) | 296763 (59.2) |
| | Occasionally | 3565 (18.3) | 100302 (20.0) |
| | Everyday | 3761 (19.3) | 104269 (20.8) |
| Use of lipid-lowering agents | Yes | 6818 (28.6) | 139738 (23.8) |
| Current smoker | Yes | 2392 (10.0) | 69971 (11.9) |
| Walking or physical exercise > 1 h / wk | Yes | 10510 (49.6) | 267136 (50.5) |
| Increase in weight > 10 kg since age of 20 y | Yes | 7348 (34.6) | 154621 (29.2) |
| Estimated GFR | mL/min/1.73 m$^2$ | 66.42 (16.04) | 69.47 (15.75) |
| BMI | kg/m$^2$ | 23.15 (3.55) | 22.63 (3.43) |
| GGT | U/L | 38.98 (66.47) | 32.55 (45.30) |
| AST | U/L | 25.66 (13.80) | 24.05 (10.57) |
| ALT | U/L | 21.59 (16.54) | 20.17 (13.26) |
| HbA1c | % | 5.79 (0.73) | 5.74 (0.70) |
| LDL cholesterol | mg/dL | 120.76 (30.90) | 124.02 (31.30) |
| Systolic blood pressure | mmHg | 130.70 (16.59) | 129.38 (17.36) |
| Triglycerides | mg/dL | 118.60 (70.72) | 113.92 (72.42) |
| Uric acid | mg/dL | 5.35 (1.34) | 5.22 (1.36) |

ALT, alanine aminotransferase; AST, aspartate aminotransferase; BMI, body mass index; GFR, glomerular filtration rate; GGT, gamma-glutamyl transpeptidase; HbA1c, hemoglobin A1c; HDL, high-density lipoprotein; LDL, low-density lipoprotein.

**Table 2. Results of univariable Cox model analysis.**

| Variable (reference) | Category or unit | Univariable Cox model | | |
|---|---|---|---|---|
| | | HR | 95% CI | p value |
| Sex (women) | Men | 1.21 | 1.18–1.24 | <0.001 |
| Age (0 to < 40 years) | 40 to < 50 years | 1.36 | 0.99–1.86 | 0.090 |
| | 50 to < 60 years | 1.98 | 1.45–2.71 | <0.001 |
| | 60 to < 70 years | 2.62 | 1.93–3.56 | <0.001 |
| | 70 to < 80 years | 3.65 | 2.69–4.96 | <0.001 |
| | 80 years and over | 4.44 | 3.27–6.04 | <0.001 |
| Cerebrovascular disease (absence) | Presence | 1.49 | 1.44–1.54 | <0.001 |
| Any malignancy (absence) | Presence | 1.54 | 1.47–1.60 | <0.001 |
| Dementia (absence) | Presence | 1.60 | 1.49–1.72 | <0.001 |
| Myocardial infarction (absence) | Presence | 1.52 | 1.39–1.67 | <0.001 |
| Renal disease (absence) | Presence | 1.61 | 1.49–1.74 | <0.001 |
| Rheumatic disease (absence) | Presence | 1.34 | 1.24–1.44 | <0.001 |
| Liver disease (absence) | Presence | 2.03 | 1.52–2.70 | <0.001 |
| Congestive heart failure (absence) | Presence | 1.65 | 1.58–1.71 | <0.001 |
| Chronic pulmonary disease (absence) | Presence | 1.29 | 1.25–1.33 | <0.001 |
| Diabetes (absence) | Presence | 1.50 | 1.42–1.58 | <0.001 |
| Hypertension (absence) | Presence | 1.51 | 1.47–1.54 | <0.001 |
| Pylori-infected gastritis (absence) | Presence | 1.29 | 1.22–1.36 | <0.001 |
| History of gastrectomy (absence) | Presence | 2.22 | 1.40–3.52 | <0.001 |
| HIV infection (absence) | Presence | 0.57 | 0.08–4.06 | 0.576 |
| Crohn's disease (absence) | Presence | 1.61 | 0.98–2.62 | 0.058 |
| Spinal injury (absence) | Presence | 1.60 | 0.23–11.37 | 0.637 |
| Total parenteral nutrition (absence) | Presence | NE | NE | NE |
| Frequency of alcohol consumption | Never | 1.00 | | |
| | Occasionally | 0.90 | 0.87–0.93 | <0.001 |
| | Everyday | 0.91 | 0.88–0.95 | <0.001 |
| Use of lipid-lowering agents (no) | Yes | 0.83 | 0.81–0.86 | <0.001 |
| Current smoker (no) | Yes | 1.09 | 1.04–1.14 | <0.001 |
| Walking or physical exercise > 1 h / wk (no) | Yes | 0.91 | 0.88–0.93 | <0.001 |
| Increase in weight > 10 kg since age of 20 y (no) | Yes | 1.33 | 1.29–1.37 | <0.001 |
| Estimated GFR | 1 mL/min/1.73 m$^2$ | 0.99 | 0.99–0.99 | <0.001 |
| BMI | 1 kg/m$^2$ | 1.05 | 1.04–1.05 | <0.001 |
| GGT | 100 U/L | 1.15 | 1.14–1.16 | <0.001 |
| AST | 10 U/L | 1.07 | 1.06–1.07 | <0.001 |
| ALT | 10 U/L | 1.05 | 1.05–1.06 | <0.001 |
| HbA1c | 1% | 1.06 | 1.05–1.07 | <0.001 |
| LDL cholesterol | 10 mg/dL | 0.96 | 0.96–0.97 | <0.001 |
| Systolic blood pressure | 10 mmHg | 1.04 | 1.03–1.05 | <0.001 |
| Triglycerides | 100 mg/dL | 1.10 | 1.08–1.12 | <0.001 |
| Uric acid | 1 mg/dL | 1.07 | 1.06–1.07 | <0.001 |

ALT, alanine aminotransferase; AST, aspartate aminotransferase; BMI, body mass index; CI, confidence interval; GFR, glomerular filtration rate; GGT, gamma-glutamyl transpeptidase; HbA1c, hemoglobin A1c; HR, hazard ratio; LDL, low-density lipoprotein.

Unlike previous reports, a higher prevalence of gallstones among men was observed in this study. Several classic epidemiologic studies have reported that pregnancy and female sex hormones place women at a higher risk [3–6]. The high prevalence of gallstones among men in

**Table 3. Results of multivariable Cox model analysis.**

| Variable (reference), n = 541,972, events = 20,810 | Category or unit | Multivariable Cox model | | |
|---|---|---|---|---|
| | | HR | 95% CI | p value |
| Sex (women) | Men | 1.09 | 1.06–1.12 | <0.001 |
| Age (0 to < 40 years) | 40 to < 50 years | 1.31 | 0.95–1.81 | 0.103 |
| | 50 to < 60 years | 1.94 | 1.41–2.68 | <0.001 |
| | 60 to < 70 years | 2.54 | 1.85–3.48 | <0.001 |
| | 70 to < 80 years | 3.34 | 2.44–4.59 | <0.001 |
| | 80 years and over | 3.97 | 2.89–5.45 | <0.001 |
| Cerebrovascular disease (absence) | Presence | 1.14 | 1.10–1.19 | <0.001 |
| Any malignancy (absence) | Presence | 1.30 | 1.24–1.36 | <0.001 |
| Dementia (absence) | Presence | 1.13 | 1.04–1.23 | 0.005 |
| Myocardial infarction (absence) | Presence | 1.02 | 0.92–1.14 | 0.647 |
| Renal disease (absence) | Presence | 1.14 | 1.05–1.24 | 0.003 |
| Rheumatic disease (absence) | Presence | 1.23 | 1.14–1.33 | <0.001 |
| Liver disease (absence) | Presence | 1.56 | 1.16–2.10 | 0.003 |
| Congestive heart failure (absence) | Presence | 1.17 | 1.12–1.23 | <0.001 |
| Chronic pulmonary disease (absence) | Presence | 1.15 | 1.12–1.19 | <0.001 |
| Diabetes (absence) | Presence | 1.15 | 1.08–1.22 | <0.001 |
| Hypertension (absence) | Presence | 1.10 | 1.06–1.13 | <0.001 |
| Pylori-infected gastritis (no) | Yes | 1.25 | 1.18–1.32 | <0.001 |
| History of gastrectomy (no) | Yes | 1.46 | 0.88–2.43 | 0.142 |
| Use of lipid-lowering agents (no) | Yes | 1.02 | 0.99–1.06 | 0.159 |
| Current smoker (no) | Yes | 1.04 | 0.99–1.09 | 0.104 |
| Walking or physical exercise > 1 h / wk (no) | Yes | 0.91 | 0.94–0.89 | <0.001 |
| BMI | 1 kg/m$^2$ | 1.04 | 1.04–1.05 | <0.001 |
| GGT | 100 U/L | 1.15 | 1.13–1.16 | <0.001 |
| HbA1c | 1% | 1.03 | 1.01–1.04 | 0.001 |
| LDL cholesterol | 10 mg/dL | 0.98 | 0.98–0.99 | <0.001 |
| Systolic blood pressure | 10 mm Hg | 0.98 | 0.97–0.99 | <0.001 |
| Triglycerides | 100 mg/dL | 1.04 | 1.02–1.06 | <0.001 |

BMI, body mass index; CI, confidence interval; GGT, gamma-glutamyl transpeptidase; HbA1c, hemoglobin A1c; HR, hazard ratio; LDL, low-density lipoprotein.

our study population is an indicator of a significant shift in the epidemiology of gallstone disease. One possible explanation for this is that the obesity rate among men has been on an upward trend in Japan since 2013. In 2019, 33.0% of men were considered overweight due to having an average body mass index of 25 or greater, 4.4% higher than in 2013. For women, the overall average was 22.3%, up 2.0% from 2013. A smaller proportion of women were obese compared to men across all age groups in 2019 [24]. Similarly, Sung et al. reported that the prevalence of asymptomatic cholelithiasis was higher in men than women over the age of 50 in Japan in 2017, concluding that this was due to the higher BMI, triglyceride, and fasting plasma glucose levels found in men than in women [25].

It had previously been suggested that patients with rheumatoid disease may be at increased risk for gallstones, but evidence for this association had been lacking before the present study. Our results are consistent with the suggestions of several past reports. Chen demonstrated a significant inverse relationship between systemic chronic inflammation and dyslipidemia in patients with rheumatoid disease [26]. In a study of 224 rheumatoid arthritis patients in Japan, Ito et al. found a significantly higher incidence of gallstones than in the general population,

concluding that the chronic inflammatory state due to the condition altered lipid profile and gallbladder function, leading to the higher incidence [27]. In addition, other conditions such as low levels of physical activity or use of steroidal treatments by those with rheumatoid arthritis may be associated with an increased incidence of gallstones.

Among the few previous reports on the relationship between malignant disease and the development of cholelithiasis, Reimar W. et al. [28] found an association similar to the one found in the present study. Their population-based cohort study of 51,228 cancer patients reported that biliary obstruction due to malignant metastases to the liver in cancer patients, together with malnutrition and rapid weight loss in patients with malignant tumors, lead to a higher incidence of cholecystitis than in the general population. Thus, the specific conditions of patients with malignant diseases such as systemic inflammation and abnormal lipid metabolism appear to contribute to the development of cholelithiasis.

Interestingly, the present study found that *H. pylori* infection was positively correlated with the development of cholelithiasis. In a cross-sectional study, Takahashi et al. [29] reported that *H. pylori* infection was positively correlated with the presence of gallstones. *H. pylori* has proven to be a major pathogen in gastric diseases such as chronic gastritis, gastric ulcers, duodenal ulcers, and gastric cancer [30]. Furthermore, *H. pylori* infection has also been reported to be associated with extra-gastric diseases such as dyslipidemia, type 2 diabetes, insulin resistance, metabolic syndrome, and increased bilirubin levels [31–35]. Given that *H. pylori* infection is related to these metabolic diseases, it may also be an indirect mediator of the gallbladder environment, leading to the development of gallstones.

This study also confirms that chronic lung disease with smoking as a risk factor increases the risk of cholelithiasis. There are several reports indicating that smoking is a risk factor for developing the disease [36, 37], and the present study identifies chronic lung disease as a risk factor for it. However, little is known about the biological mechanisms by which these factors may cause the disease. One possibility is that the dozens of toxic compounds contained in tobacco smoke [37, 38] may have a detrimental effect on the gallbladder. The relevant biological mechanisms need further elucidation.

This study also found a higher incidence of cholelithiasis in patients with chronic kidney disease. Several previous reports have also shown an association between renal disease and cholelithiasis [39]. Dysautonomia, which is common in patients with renal failure, impairs gallbladder motility, and renal failure increases biliary cholesterol and decreases primary bile acids in bile, making such patients more susceptible to cholelithiasis [40]. Thus, previous research found that several abnormalities related to renal function appear to promote gallstone formation, a finding consistent with that of the present study.

In addition, the present study found an association between gallstones and dementia. Miyamoto et al [41] found in a cross-sectional study that biliary sludge developed in older adults with dementia, which may have been due to the poor nutritional intake, immobility, and frailty commonly observed in dementia patients. This suggests that the development of cholelithiasis may be influenced by complex factors such as low levels of nutritional intake and physical activity.

Previous studies have also found an association between gallstones and hypertension [42]. It is important to note that the present study found two seemingly contradictory associations in relation to hypertension (Table 3): simultaneous treatment for hypertension in the insurance record was associated with an increased risk of cholelithiasis (HR: 1.10), while higher SBP in the medical record was associated with a decreased risk of the disease (HR: 0.98). This apparent contradiction is explained by the fact that most patients being treated for hypertension can be expected to have been prescribed anti-hypertensive drugs, and therefore to have had blood pressure in the normal range when they were treated for gallstones.

This study also compared the characteristics of individuals with gallstones who received treatment for the disease with those that did not receive treatment for it, and found that individuals who had comorbid lifestyle-related diseases were more likely to receive treatment for gallstones (S2 Table).

Additionally, this study found trends for several factors that differ from those of previous reports. It did not find associations between the development of cholelithiasis and parenteral nutrition, spinal cord injury, or gastrectomy, likely due to the small number of cases involving these factors.

The major strength of this study is its presentation of incidence trends in a recent 7.5-year period based on a large sample. However, this study also has several limitations. First, the incidence of gallstones was defined according to the International Classification of Diseases (Tenth Edition) coding, which has not yet been validated using medical record data. Secondly, age at onset could not be accurately ascertained, so age at diagnosis was used as a proxy. Thirdly, in the health check-up, the questionnaire items did not accurately capture data on frequency and quantity of alcohol consumption, and these variables could not be assessed in a multivariable regression analysis. Finally, the study confirmed that enrollees with newly-diagnosed gallstones did not have gallstones during the one-year covariate assessment window. However, if a patient who already had gallstones did not use insurance for gallstone treatment during the baseline period, he or she may have been treated as a new onset patient. The definition of cholelithiasis used in this study includes patients who made an insurance claim for cholelithiasis at least once during the observation period, but patients with cholelithiasis who did not visit a healthcare provider could not be included in the study. Despite these limitations, this study provides novel evidence for the association between gallstone disease and a large number of risk factors.

## Conclusion

This study reveals that the risk of cholelithiasis in the general Japanese population is increased by male sex, cerebrovascular disease, malignancy, dementia, rheumatic disease, chronic pulmonary disease, hypertension, and *H. pylori* gastritis. These findings may contribute to efforts to reducing the incidence of cholelithiasis.

## Supporting information

**S1 Table. Insurance claim codes for gallstone-related medical interventions.**
(DOCX)

**S2 Table. Characteristics of individuals with gallstones who underwent treatment for the disease and those who did not undergo treatment.** ALT, alanine aminotransferase; AST, aspartate aminotransferase; BMI, body mass index; GFR, glomerular filtration rate; GGT, gamma-glutamyl transpeptidase; HbA1c, hemoglobin A1c; HDL, high-density lipoprotein; LDL, low-density lipoprotein.
(DOCX)

**S3 Table. Spearman's correlation matrix.** ALT, Alanine aminotransferase; AST, aspartate aminotransferase; BMI, body mass index; GFR, glomerular filtration rate; GGT, gamma-glutamyl transpeptidase; HbA1c, hemoglobin A1c; LDL, low-density lipoprotein.
(DOCX)

## Acknowledgments

We thank Professor Yoshiki Miyachi of Kyoto University and Shizuoka Graduate University of Public Health for his useful comments and criticism.

## Author Contributions

**Conceptualization:** Kazuya Higashizono, Eiji Nakatani.

**Data curation:** Eiji Nakatani.

**Formal analysis:** Kazuya Higashizono.

**Funding acquisition:** Eiji Nakatani.

**Investigation:** Kazuya Higashizono, Philip Hawke, Shuhei Fujimoto, Noriyuki Oba.

**Methodology:** Kazuya Higashizono, Eiji Nakatani.

**Project administration:** Kazuya Higashizono.

**Resources:** Eiji Nakatani.

**Software:** Eiji Nakatani.

**Supervision:** Eiji Nakatani.

**Validation:** Kazuya Higashizono.

**Visualization:** Kazuya Higashizono, Philip Hawke.

**Writing – original draft:** Kazuya Higashizono.

**Writing – review & editing:** Kazuya Higashizono, Eiji Nakatani, Philip Hawke, Shuhei Fujimoto, Noriyuki Oba.

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
