## [Decision Letter · Decision Letter 0]

29 Sep 2022

PONE-D-22-24368Risk factors for gallstone disease onset in Japan: findings from the Shizuoka Study, a population-based cohort studyPLOS ONE

Dear Dr. Higashizono,   

Thank you for submitting your manuscript to PLOS ONE. After careful consideration, we feel that it has merit but does not fully meet PLOS ONE’s publication criteria as it currently stands. Therefore, we invite you to submit a revised version of the manuscript that addresses the points raised during the review process.

We look forward to receiving your revised manuscript.

Kind regards,

Donovan Anthony McGrowder, PhD., MA., MSc

Academic Editor

PLOS ONE

Additional Editor Comments:

Dear Dr. Higashizono,   

 Your manuscript “Risk factors for gallstone disease onset in Japan: findings from the Shizuoka Study, a population-based cohort study” has been assessed by our reviewers. They have raised a number of points which we believe would improve the manuscript and may allow a revised version to be published in PLOS ONE. Their reports, together with any other comments, are below.

 If you are able to fully address these points, we would encourage you to submit a revised manuscript to PLOS ONE.

 Regards,

Dr. Donovan McGrowder

Associate Editor

Reviewers' comments:

Reviewer's Responses to Questions

**Comments to the Author**

1. Is the manuscript technically sound, and do the data support the conclusions?

Reviewer #1: Yes

Reviewer #2: Yes

2. Has the statistical analysis been performed appropriately and rigorously? 

Reviewer #1: Yes

Reviewer #2: Yes

3. Have the authors made all data underlying the findings in their manuscript fully available?

Reviewer #1: No

Reviewer #2: Yes

4. Is the manuscript presented in an intelligible fashion and written in standard English?

Reviewer #1: Yes

Reviewer #2: Yes

5. Review Comments to the Author

Reviewer #1: General comments:

The authors investigated the risk factors for gallstones disease onset in Japan via two large National Health Insurance System and National Health Insurance System and the Medical Care System for Elderly in the Latter Stage of Life in Shizuoka. The authors found the risk factors of increased gallstones disease included male sex, cerebrovascular disease, any malignancy, dementia, rheumatic disease, chronic pulmonary disease, hypertension, and H. pylori-infected gastritis.

Minor comments:

1. According to the Table 3, presence of the hypertension increased the risk of the gallstones disease with hazard ratio (HR) 1.10 (1.06-1.13), p<0.001. However, in systolic blood pressure (SBP), higher SBP reduced the risk of gallstones disease (HR: 0.98, 95% confidence interval: 0.97-0.99, p<0.001). Among this discrepancy, would author offer the explanation or discussion for it?

2. May we kindly ask for the definition of the liver disease? Moreover, in Table 2 and Table 3, the parameters shown are not the same. For example, there were AST/ALT/uric acid/eGFR in Table 2 but not in Table 3. Is that because the statistic insignificance? But you showed some parameters, such as history of gastrectomy or use of lipid-lowering agents without significant also in Table 3. It would be grateful if you would explain it.

Reviewer #2: In this manuscript “Risk factors for gallstone disease onset in Japan: findings from the Shizuoka Study, a population-based cohort study”, Dr. Higashizono and colleagues carried out the large population-based cohort studies in Japanese population about gallstone onset. The authors suggest some new risk factors of cholelithiasis, such as male, dementia, rheumatic disease, chronic pulmonary disease, hypertension, and H. pylori gastritis. I believe the research findings bring clinical benefit in cholelithiasis prevention and treatment. Therefore, I recommend the authors considering the following minor points.

1. Methods: Please describe the detail methods for detecting gallstones. Where did the medical checks take place? What were the gallstone detecting devices? Especially, please clarify the methods to distinguish gallstone from other gallbladder diseases such as polyps and segmental adenomyomatosis.

2. Results, Gallstone disease onset: Please discuss about newly detection rate of gallstones by comparing with those of former reports.

3. Results, Gallstone disease onset: The authors showed that approximately 20% of gallstone disease patients had undergone medical intervension. The morbidity seems to be considerably high. For example, Morris-Stiff et al. had studied the patients with asymptomatic gallstone and reported that the gallstone related symptoms developed at approximately 2% per year (Clin Gastroenterol Hepatol. 2022). Please discuss about the difference of the onset rates.

4. Results, Gallstone disease onset: Please compare and discuss the demographical and clinical data between the patient groups, who received and did not receive medical intervention for gallstone disease.

5. Table 3: Please indicate and discuss the result of frequency of alcohol consumption in multivariable Cox model analysis.

6. I also recommend the authors examining the effect of alcohol consumption quantity for gallstone forming.

7. I am very interested in the natural history of cholelithiasis in this population. Do the authors have any additional follow up data after initial gallstone detection?

8. Supporting information: the font size of the table is too small to read easily. Please use larger letters.

---

## [Author Response · Author response to Decision Letter 0]

23 Nov 2022

Responses to the comments of Reviewer #1

We wish to express our appreciation to the reviewer for the insightful comments, which have helped us significantly improve our paper.

1. According to the Table 3, presence of the hypertension increased the risk of the gallstones disease with hazard ratio (HR) 1.10 (1.06-1.13), p<0.001. However, in systolic blood pressure (SBP), higher SBP reduced the risk of gallstones disease (HR: 0.98, 95% confidence interval: 0.97-0.99, p<0.001). Among this discrepancy, would author offer the explanation or discussion for it?

Response: 

We greatly appreciate this valuable comment. In accordance with the suggestion, we have made the following revision to the Discussion (lines 296–307):

“It is important to note that this study found two seemingly contradictory associations in relation to hypertension (Table 3): simultaneous treatment for hypertension in the insurance record was associated with an increased risk of cholelithiasis (HR: 1.10), while higher SBP in the medical record was associated with a decreased risk of the disease (HR: 0.98). This apparent contradiction is explained by the fact that most patients being treated for hypertension can be expected to have been prescribed anti-hypertensive drugs, and therefore to have had blood pressure in the normal range when they were treated for gallstones.”

2. May we kindly ask for the definition of the liver disease? Moreover, in Table 2 and Table 3, the parameters shown are not the same. For example, there were AST/ALT/uric acid/eGFR in Table 2 but not in Table 3. Is that because the statistic insignificance? But you showed some parameters, such as history of gastrectomy or use of lipid-lowering agents without significant also in Table 3. It would be grateful if you would explain it.

Response: 

We are grateful for the insightful comment. In accordance with your suggestions, we have added the text below to the statistical analysis section methods section (lines 136–139) and respond below. 

“To carry out the analysis conservatively, no model selection was carried out, and all variables that reached statistical significance in the univariable regression analysis were entered into the multivariable regression analysis.”

In Supplementary Table 1, AST and GGT, ALT and GGT, and eGFR and age, were also correlated. Therefore, AST, ALT and eGFR were not fed into the multivariate model. In other words, GGT and age were included in the multivariate model. Also, the reason for including non-significant variables such as a history of gastrectomy and use of lipid-lowering agents in the multivariate model is that the model selection method was not implemented conservatively (such that only significant variables were left), and variables that were significant in the univariate analysis were simply put into the multivariate model. 

Responses to the comments of Reviewer #2

We wish to express our appreciation to the reviewer for the insightful comments, which have helped us significantly improve our paper.

1. 1. Methods: Please describe the detail methods for detecting gallstones. Where did the medical checks take place? What were the gallstone detecting devices? Especially, please clarify the methods to distinguish gallstone from other gallbladder diseases such as polyps and segmental adenomyomatosis.

Response: 

We greatly appreciate the valuable comments. The onset of gallstones is defined according to previous report [19], and our analysis was carried out using the same method as that report. Additionaly, only diagnoses of cholelithiasis entered as "no suspicion" were extracted from the medical claims data. On the other hand, from the insurance claim data alone, it was unclear what instruments were used to detect gallstones and how gallstones were differentiated from other gallbladder diseases (e.g., polyps and segmental adenomyomatosis).

2. 2. Results, Gallstone disease onset: 

Please discuss about newly detection rate of gallstones by comparing with those of former reports.

Response: 

We greatly appreciate the insightful comment. To our knowledge, there are no reports on the incidence (not prevalence) of gallstones in Japan. Therefore, we believe that it is difficult to compare the incidence in our study with others.

3. 3. Results, Gallstone disease onset: The authors showed that approximately 20% of gallstone disease patients had undergone medical intervension. The morbidity seems to be considerably high. For example, Morris-Stiff et al. had studied the patients with asymptomatic gallstone and reported that the gallstone related symptoms developed at approximately 2% per year (Clin Gastroenterol Hepatol. 2022). Please discuss about the difference of the onset rates.

Response: 

We very much appreciate the valuable comment. The population in this study was individuals diagnosed with gallstones in the claim database. Thus, we cannot know whether these patients were symptomatic or not. The incidence of symptomatic cholelithiasis (cholecystitis) was not the focus on our study, and therefore, the nature of the asymptomatic cholelithiasis reported in the Morris-Stiff et al. study (Clin Gastroenterol Hepatol, 2022) cannot be compared to the results our study.

4. 4. Results, Gallstone disease onset: Please compare and discuss the demographical and clinical data between the patient groups, who received and did not receive medical intervention for gallstone disease.

Response: 

We greatly appreciate the reviewer’s insightful comment. In accordance with the suggestion, we have made the following revision to the Methods, Result, Discussion and S1 Table.

In Methods:

“Insurance claim codes of for gallstone-related medical interventions.” (lines 115-116)

In Results:

“The baseline characteristics of individuals with gallstones who underwent treatment for the disease and those who did not are shown in S1 Table.” (lines 170-172)

In Discussion:

“This study also compared the characteristics of individuals with gallstones who received treatment for the disease with those that did not receive treatment for it, and found that individuals who had comorbid lifestyle-related diseases were more likely to receive treatment for gallstones (S2 Table).” (lines 304-307)

5. 5. Table 3: Please indicate and discuss the result of frequency of alcohol consumption in multivariable Cox model analysis.

6. I also recommend the authors examining the effect of alcohol consumption quantity for gallstone forming.

Response: 

In the health check-up data, the questionnaire on frequency and quantity of alcohol consumption generated in relation to each other. However, when analyzed in combination, it is difficult to calculate the amount of alcohol consumed by the subjects. 

Item Answer

How often do you drink alcohol? (Sake, shochu, beer, wine, whiskey, brandy, etc.) 1. Everyday

2. Sometimes

3. Rarely drink（Cannot drink）

How much do you drink per day? Sake (180ml), middle-size beer (500ml), shochu (80ml), whiskey (60ml), two glasses of wine (240ml)? 1. Less than 180ml

2. Over180 ml, less than 360 ml

3. Over 360 ml, less than 540 ml

4. Over 540 ml

The data in Table 3 was the minimal amount required for the analysis of frequency of alcohol consumption, but there were also many missing measurements, so multivariate analysis could not be used. For these reasons, it is difficult to perform the analysis suggested by the reviewer. This point was added to the limitations as follows. (lines 317-319).

“Thirdly, in the health check-up, the questionnaire items did not accurately capture data on frequency and quantity of alcohol consumption, and these variables could not be assessed in a multivariable regression analysis.”

6. 7. I am very interested in the natural history of cholelithiasis in this population. Do the authors have any additional follow up data after initial gallstone detection?

Response: We very much appreciate the suggestion. It would be possible to compile additional data on follow-up after the initial discovery of gallstones by extracting it from a large data set. However, we would prefer to refrain from reporting on this issue in this manuscript, as doing to would produce a prognostic study after the onset of the disease rather than a risk exploration study up to the onset of the disease, which is the focus of this study.

7. 8. Supporting information: the font size of the table is too small to read easily. Please use larger letters.

Response: We greatly appreciate the valuable comment. In accordance with it, the font size of Supplementary Table 1 has been increased and the table has been divided into two parts to make it easier to read.

We believe that the above responses fully address the reviewer’s comments.

Thank you once again to the reviewers for their comments on the paper. We hope that our revised manuscript is now suitable for publication.

---

## [Editor Report · Decision Letter 1]

14 Dec 2022

Risk factors for gallstone disease onset in Japan: findings from the Shizuoka Study, a population-based cohort study

PONE-D-22-24368R1

Dear Dr. Higashizono,

We’re pleased to inform you that your manuscript has been judged scientifically suitable for publication and will be formally accepted for publication once it meets all outstanding technical requirements.

Kind regards,

Donovan Anthony McGrowder, PhD., MA., MSc

Academic Editor

PLOS ONE

Additional Editor Comments (optional):

Dear Dr. Higashizono,

The manuscript entitled “Risk factors for gallstone disease onset in Japan: findings from the Shizuoka Study, a population-based cohort study” was revised in accordance with the reviewers’ comments and is provisionally accepted pending final checks for formatting and technical requirements.

Regards,

Dr. Donovan McGrowder (Academic Editor)

---

## [Editor Report · Acceptance letter]

21 Dec 2022

PONE-D-22-24368R1 

Risk factors for gallstone disease onset in Japan: findings from the Shizuoka Study, a population-based cohort study 

Dear Dr. Higashizono:

I'm pleased to inform you that your manuscript has been deemed suitable for publication in PLOS ONE. Congratulations! Your manuscript is now with our production department. 

Kind regards, 

on behalf of

Dr. Donovan Anthony McGrowder 

Academic Editor

PLOS ONE